# Novel Dual Endothelin Inhibitors in the Management of Resistant Hypertension

**DOI:** 10.3390/life13030806

**Published:** 2023-03-16

**Authors:** Chrysoula Boutari, Fotios Siskos

**Affiliations:** 2nd Propedeutic Department of Internal Medicine, Medical School, Aristotle University of Thessaloniki, Hippokration General Hospital, 54642 Thessaloniki, Greece

**Keywords:** resistant hypertension, cardiovascular diseases, endothelin system, endothelin receptor inhibitors, aprocitentan

## Abstract

Resistant hypertension (RH) is defined as the failure to achieve blood pressure control despite using triple combination therapy with a renin-angiotensin system inhibitor (RAS-i), a calcium antagonist, and a diuretic. The endothelin (ET) system is implicated in the regulation of vascular tone, primarily through vasoconstriction, intervenes in cardiac contractility with inotropic effects, and contributes to water and sodium renal reabsorption. ET inhibitors, currently approved for the treatment of pulmonary hypertension, seem to be also useful for essential hypertension and RH as well. Studies into the development of new dual ET inhibitors, which inhibit both type A and B ET (ET_A_ and ET_B_) receptors, present initial results of managing RH. Aprocitentan (ACT-132577) is a novel, orally active and well tolerated dual ET receptor antagonist, which has been examined in several experimental studies and clinical trials with promising results for RH control. The recent publication of the large PRECISION study in *The Lancet* journal provides further reassurance regarding the efficacy and safety of aprocitentan for RH, with the aim of overcoming unmet needs in the management of this difficult group of patients.

## 1. Introduction

Hypertension is the most common chronic disease globally and represents a major risk factor for cardiovascular disease and death. Several clinical studies in the past have proved the benefits of reducing blood pressure (BP) values within normal limits in hypertensive patients. Despite the progress in the management of hypertension during recent years, and the increased awareness of the importance of its prevention, the prevalence of hypertension remains high globally [1]. In a recent pooled analysis of 1201 representative-population studies, published in 2021, including 104 million patients from 184 countries, representing almost 99% of the global population, the prevalence of hypertension in adults aged 30–79 years was estimated at 34% for men and 32% for women. The rates for hypertensive patients who received treatment was 38% and 47% for men and women, respectively. Moreover, among patients who received an anti-hypertensive regimen almost half of them managed to achieve BP control [2].

Factors that can contribute to poor BP control include patients’ non-adherence to treatment or to lifestyle modifications, clinical inertia, or physicians’ non-adherence to guidelines [3]. In recent times, resistant hypertension (RH) is recognized at an increasing rate as a substantial factor accounting for inadequate BP control among hypertensives, constituting an important public health issue. RH is associated with the increased risk of adverse outcomes and increased cardiovascular morbidity and mortality compared with non-RH, indicating how significant its management is in clinical practice [4,5]. A crucial parameter of the management of RH is the identification and subsequent treatment of potentially reversible causes of secondary hypertension, such as obstructive sleep apnea (OSA), primary aldosteronism, and renal artery stenosis. Identifying the secondary cause of RH is important in the RH management, frequently differs from the treatment of primary hypertension, and depends upon the specific disorder [6]. Additionally, intensification of lifestyle interventions and pharmaceutical treatment, in addition to other anti-hypertensive drugs with different mechanisms of action, are some options in the management of RH. In addition, non-pharmaceutical methods have been tested, such as renal denervation or carotid baroreceptor stimulation, with promising results; however, further evaluation is needed [5,7].

The identification of the role of the endothelin (ET) system in vascular tone and blood pressure increase, and its potential involvement in several cardiovascular diseases, resulted in the development of ET receptor inhibitors and, subsequently, in the beginning of a new era in pharmacology [8]. Until now, ET inhibitors have been approved only for the treatment of pulmonary hypertension, but an increasing number of experimental and clinical studies have been conducted evaluating the use of ET inhibitors in several diseases, including essential hypertension and RH. The development of new dual ET inhibitors, which inhibit both type A and B ET (ET_A_ and ET_B_) receptors, provides improved efficacy of this pharmacological category, with promising results in controlling RH [9,10,11] [Figure 1].

The purpose of this review is to present recent data regarding the development of novel dual ET inhibitors and their position in the management of RH through experimental studies and clinical trials, highlighting simultaneously the importance of early recognition and efficient management of RH.

## 2. Resistant Hypertension

Resistant hypertension (RH) is defined as BP levels over the target limits despite the simultaneous use of three anti-hypertensive drugs from different pharmacological classes, ideally including a long-acting calcium channel blocker, a renin-angiotensin system inhibitor (RAS-i), and a diuretic administered at maximum doses or maximum tolerated doses [12]. Furthermore, there is a growing body of evidence showing the benefits of mineralocorticoid receptor antagonists, such as eplerenone and spironolactone, for improving BP control in patients with RH, regardless of circulating aldosterone levels. Therefore, this class of drugs should be considered for patients whose BP remains high after treatment with three antihypertensive drugs to maximal or near maximal doses [6]. Patients who manage to control their BP with the use of four or more antihypertensive drugs are also considered to have RH. Exclusion of the ‘white-coat effect’ and patients’ non-adherence to an antihypertensive regimen are both required in order to make a diagnosis of true RH. If any of the aforementioned conditions are present, the RH is characterized as pseudo-resistant, and when the existence of pseudo-RH cannot be excluded then it is referred to as apparent-RH. A prerequisite for all of the above is the utilization of a proper BP measurement technique by a physician [12,13].

According to a recent systematic review and meta-analysis, the worldwide prevalence of true-RH in the general population of treated hypertensive patients is 10.3%, while the prevalence of apparent- and pseudo-RH are 14.7% and 10.3%, respectively [14]. These results are in accordance with those from a previous meta-analysis, in which the prevalence of apparent-RH was estimated to be 13.7% [15], as well as in accordance with the results from other observational studies in which the prevalence fluctuates between 9% and 15% among treated hypertensive patients [4,16]. However, the prevalence of true-RH seems to be greater among specific populations with hypertension, such as patients with chronic kidney disease (CKD) in which the prevalence of true-RH was estimated to be 2 to 3 times higher in comparison with that in the general hypertensive population [14,17]. Moreover, the high prevalence of pseudo-RH emphasizes the importance of recognizing this condition, since 1 in 10 hypertensive patients can be classified incorrectly as having RH.

Defining RH and detection of patients with RH are of great importance, since these patients seem to have a greater risk for developing adverse cardiovascular events or death compared to the other hypertensive patients. A cohort study of 18,036 patients with RH determined the incidence of adverse cardiovascular events. In a median period of 3.8 years of follow-up, patients with RH presented a 47% higher risk of suffering from myocardial infraction, stroke, congestive heart failure, impaired renal function or death compared with non-RH patients [18]. In another study, a 2.9 times higher relative risk for cardiovascular disease was reported for patients with true-RH in comparison with non-resistant patients [19]. Moreover, target-organ damage, including carotid intima-media thickening, left ventricular hypertrophy, albuminuria and retinopathy, is more likely to appear in patients with RH among hypertensives [20].

RH manifestation is associated with the male sex, older age, black race, obesity, diabetes mellitus and CKD [21,22]. Excessive sodium intake is a common condition responsible for poor BP control despite optimal treatment, especially in western societies. Furthermore, excessive alcohol consumption can contribute to RH. Other important factors that cause BP elevation include obstructive sleep apnea, and drugs such as non-steroidal anti-inflammatories, contraceptives, cyclosporin, erythropoietin, sympathomimetics and steroids [23,24]. It would be remiss not to mention the secondary causes of hypertension, of which primary hyperaldosteronism and chronic kidney disease are the most common and may be responsible for RH [25].

Therapeutic options for RH include lifestyle interventions in combination with pharmaceutical agents. Among other lifestyle modifications, salt restriction is the most important, while volume expansion constitutes a common contributor to RH. Adding a more potent diuretic, such as chlorthalidone, may be an option, while there are clinical studies supporting the idea that the replacement of hydrochlorothiazide with chlorthalidone in equivalent doses might lead to beneficial effects in BP control. The PATHWAY-2 clinical trial was the first trial which compared the effect of an aldosterone antagonist with the non-diuretic drugs bisoprolol and doxazosin as an add-on therapy for RH. In this trial, spironolactone was superior to the other two drugs and the placebo for BP reduction, which reflects the important role of sodium retention in RH, as highlighted by the authors [26]. Alternative drugs for RH management include direct vasodilators, such as minoxidil and hydralazine, as well as central a_2_ agonists such as clonidine [13,27].

Renal denervation and carotid baroreceptor stimulation are still under investigation. Simplicity HTN-3 was the first randomized, single-blinded trial evaluating renal nerve ablation in drug-resistant hypertensives; however, this did not have the expected results. More specifically, 553 patients with RH were randomized to undergo renal denervation or a sham procedure. Within both groups of patients, the reduction of BP was significantly decreased in comparison with baseline levels, but there was no statistical significant difference in BP reduction between the two groups [28]. However, the disappointment created by the findings of this study was counterbalanced by the recent findings of several randomized controlled trials (RCTs) that confirmed the efficacy and safety of renal denervation [29]. Despite all the above-mentioned available therapies, RH remains an important health issue. The need for new therapies for RH seems to be imperative to efficiently handling this condition.

## 3. Endothelin System in Cardiovascular Disease

The ET system comprises three isoforms of the ET peptide, each of which is expressed predominantly in different tissues. Endothelin-1 (ET-1) is the most plentiful ET in the cardiovascular system and is considered to be the most potent and long-acting vasoconstrictor in the human body, exerting its action through activation of ET receptors. Two types of ET receptors have been recognized, type A (ET_A_) and type B (ET_B_), which are G-protein coupled receptors and have equipotent affinity to ET-1 [30]. These receptors are present in several cell types, such as smooth muscle cells, endothelial cells, cardiomyocytes, fibroblasts, macrophages, glomeruli and tubular cells [30,31].

ET-1 is also expressed in many types of cells, principally in vascular endothelial cells. Its production is regulated by several stimuli, including, among others, hypoxia, tumor necrosis factor, shear stress, insulin, norepinephrine, angiotensin II, nitric oxide, natriuretic peptides and free radicals. Synthesis of mature, active ET-1 requires the action of two ET converting enzymes, which participate in the modification of an inactive precursor, called big-ET [32]. ET-1 seems to act in a paracrine and autocrine manner, which is also reflected in the proportionally low levels of ET-1 concentration in the blood circulation [33].

The main role of ET-1 in the cardiovascular system is the regulation of vascular tone, primarily through vasoconstriction. However, it also seems to intervene in additional processes, such as cardiac contractility, in which it exerts inotropic effects, and renal water and sodium reabsorption; in addition, it participates in mitogenic, inflammatory and apoptotic processes. All the above attributes of the ET system contribute to a potentially central role in the pathogenesis and pathophysiology of various cardiovascular diseases, including hypertension, atherosclerosis, heart failure, pulmonary hypertension, coronary artery disease and renal disease [31,34].

Vascular smooth muscle cells express both ET_A_ and ET_B_ receptors, while endothelial cells express only ET_B_ receptors. ET-1 can cause both vasoconstriction, through the activation of ET_A_ and ET_B_ receptors on smooth muscle cells, and vasodilation, through activation of ET_B_ receptors on endothelial cells; the latter is mediated by the subsequent production of vasodilatory agents from endothelial cells, mainly nitric oxide [32,35]. However, the central role of ET_B_ receptors on endothelial cells is in the clearance of circulating ET-1. The effect of ET-1 is determined by the distribution of its receptors on different vessels [36].

Overexpression of ET-1 has been linked with several cardiovascular diseases. Firstly, in patients with arterial hypertension ET-1 levels are elevated. Despite the regulation of vascular tone, ET-1 seems to be involved in the pathogenesis of arterial hypertension through additional pathways, including the modulation of endothelial dysfunction and arterial stiffening [37]. Moreover, in pulmonary hypertension both increased expression and increased circulation of ET-1 have been demonstrated. The levels of circulating ET-1 are considered to have a prognostic value and are used as an index for the probability of hospitalization and mortality for those patients [38]. Remodeling of pulmonary vasculature is a common feature of this disease, resulting from abnormal endothelial and smooth muscle cells function, mediated by ET-1, and leads to impaired vascular tone balance and irregular cell apoptosis and proliferation control [34,37].

The ET system is also present in the heart, with the ET receptors being distributed throughout the myocardium and coronary vessels. Stimulation of ET_A_ receptors leads to positive inotropic effects in cardiomyocytes, while the stimulation of ET_B_ receptors may have the opposite effects, as evidenced by experimental models [31,39]. Myocardial hypertrophy and remodeling that occur during heart failure seem also to be mediated, at least in part, by the ET system, through the activation of inflammatory and fibrotic processes and subsequent cytokine overexpression, such as tumor necrosis factor (TNF)-a, interleukin (IL)-1 and interleukin (IL)-6 [37,40]. Furthermore, inflammatory processes involved in atherogenesis seem to be affected by the ET system. Specifically, ET-1 activates macrophages resulting in cytokine formation, and stimulates neutrophil aggregation and their adhesion to endothelial cells. Moreover, enhanced vascular smooth muscle cell proliferation, mediated by ET-1 through ETA receptors, can be added to the different ways that the ET system may participate in atherogenesis [31]. Therefore, the inflammation caused by ET-1 results both in atherosclerosis and myocardial hypertrophy, which contribute to RH [41]. In both atherosclerotic disease and in heart failure, the levels of ET-1 have been found to be increased. Importantly, increased blood levels of ET-1 after heart failure are also associated with poor outcomes and prognosis [37,42].

To summarize, the effects of the ET system on the cardiovascular system vary and sometimes may be contradictory. This depends on the type, distribution and activation of ET receptors, which may be altered in each disease state.

## 4. Endothelin Inhibitors in Hypertension

ET receptor antagonists bosentan, ambrisentan and macitentan are currently approved for the treatment of pulmonary hypertension, based upon the proposed pathogenesis of this disease, which is characterized by enhanced synthesis of ET-1 and progressive proliferation and hypertrophy of smooth muscle cells in the pulmonary vasculature. It has been suggested that these agents increase exercise capacity, improve World Health Organization (WHO) functional class, slow down WHO functional class decline, and promote favorable changes in cardiopulmonary hemodynamic variables in patients with pulmonary hypertension compared with placebo [43]. Nevertheless, the use of these drugs has been associated with a low, but appreciable, rate of serum enzyme elevations [44]. In particular, macitentan’s most common adverse effects are nasopharyngitis, headache, and anemia [45]. Ambricentan causes peripheral edema, headache, upper respiratory infections and dizziness [46]. As for the bosentan, according to the FDA labeling, in an embryo-fetal toxicity study in rats this drug showed dose-dependent teratogenic effects, such as malformation of the head, mouth, face and large blood vessels.

Bosentan was the first orally administered dual endothelin ET_A_ and ET_B_ receptor antagonist studied in hypertensive patients. A placebo-controlled, double-blind study of 293 patients with essential hypertension evaluated the effects of bosentan 4 weeks after its administration [47]. Compared to placebo, bosentan (500 or 2000 mg) resulted in a significant reduction in diastolic BP, which was equivalent to that with enalapril. However, the incidence of adverse events, such as headache, flushing, leg edema, and asymptomatic increases in serum aminotransferase levels, in the bosentan group was significantly higher than that in the other groups. Moreover, a multicenter, placebo-controlled, dose-response study evaluated the effects of darusentan, a selective ET_A_ antagonist, in patients with moderate essential hypertension after 6 weeks of treatment [48]. Although darusentan significantly reduced both diastolic and systolic BP compared to placebo, there was a trend towards more side effects, such as headaches, flushing, and peripheral edema, in the active treatment group compared to placebo group.

Subsequently, two studies on darusentan for RH followed. The DORADO trial included 379 patients with RH who were receiving three or more antihypertensive drugs, including a diuretic, at full or maximum tolerated doses, and randomized them to receive a 14 week treatment with placebo or darusentan (50 mg, 100 mg or 300 mg) once daily [49]. Darusentan resulted in a significant reduction in BP and the most frequent adverse event was fluid retention. The other study (DAR-312) was conducted on 849 patients with RH, receiving at least three BP lowering drugs, but the participants were randomized to receive the darusentan (50 mg, 100 mg or 300 mg), placebo or the central α-2 agonist guanfacine (1 mg, once daily) for 14 weeks [50]. In these patients, both office and 24 h ambulatory BP were obtained. As for the sitting office BP changes from baseline, they were not significantly different between darusentan and placebo group. In contrast, darusentan reduced mean 24 h systolic BP more than placebo or guanfacine after 14 weeks of treatment. However, due to the negative outcome concerning the sitting office BP, the company decided to withdraw the drug without further testing.

Currently, studies evaluating the potentially beneficial effects on the BP of patients with RH are conducted with aprocitentan. Aprocitentan (ACT-132577) is a novel, orally active, dual ET receptor antagonist with an ET_A_/ET_B_ inhibitory potency ratio of 1:16 and a long half-life of 44 h in humans [51,52]. It is the active metabolite of macitentan, which is a dual endothelin receptor antagonist approved for the treatment of pulmonary hypertension [45].

## 5. Novel Dual Endothelin Inhibitors

### 5.1. Pharmacokinetic-Pharmacodynamic

Aprocitentan is highly bound to plasma proteins and is excreted in both urine and feces [53]. Single- (600 mg) and multiple- (100 mg) doses’ tolerability, safety, pharmacokinetics and pharmacodynamics of aprocitentan were investigated in a first-in-human study [52]. The drug was well-tolerated across all doses. Plasma concentration-time profiles of the drug were similar after single- and multiple-dose administration. Additionally, this agent presented a half-life of 44 h and, after multiple doses and pharmacokinetic tolerability was dose-proportional. Accumulation at steady state, reached by day eight, was 3-fold. Only minor differences in exposure between healthy females and males, healthy elderly and adult individuals, fed and fasted conditions, and renal function were documented [52,53]. Of note, pharmacokinetics did not differ significantly between Japanese and Caucasian subjects [54].

### 5.2. Experimental Data

In animal studies, aprocitentan resulted in a dose-dependent decrease in BP in two animal models with hypertension: DOCA-Salt rats (low-renin model) and spontaneously hypertensive rats (SHR) (normal-renin model). This applied especially in those with low-renin characteristics. In addition, when aprocitentan was combined with renin angiotensin system (RAS) blockers, like enalapril or valsartan, a synergistic result in lowering BP was observed in both animal models [55]. Furthermore, unlike bosentan, but similar to macitentan, aprocitentan does not interfere with bile salt homeostasis and is not hepatotoxic. According to the results of studies in rodents, dual blockade of ET_A_/ET_B_ receptors seems to have a lower risk of fluid retention and vascular leakage in comparison to ET_A_-selective blockade, which causes nonselective vasodilation and vasopressin release due to overstimulation of ET_B_ receptors [56].

### 5.3. Small Clinical Studies

The safety and efficacy of aprocitentan have been established in various phase 1 and 2 studies in healthy subjects and those with hypertension as well. Besides the phase 1 studies described above, which evaluated the safety, tolerability, and pharmacokinetics of aprocitentan in healthy subjects [52,54], and in patients with renal impairment [53], another study, occasioned by the concerns for weight gain and sodium retention, evaluated the effect of three different doses (10, 25 and 50 mg) of aprocitentan on body weight and other renal and hormonal responses from baseline to day nine in healthy subjects on a high sodium diet. A total of 23 subjects with average age of 29 years completed the study. Aprocitentan induced a moderate increase in body weight without obvious dose-dependent sodium retention (mean placebo-corrected weight gain was 0.43 (90% CI 0.05–0.80), 0.77 (90% CI 0.03–1.51), and 0.83 (90% CI 0.33–1.32) kg at 10, 25 and 50 mg, respectively [57].

Since hypertension frequently co-exists with renal impairment, the tolerability and pharmacokinetics of aprocitentan in single doses have been evaluated in subjects with severe renal function impairment (SRFI) in comparison to matched healthy subjects. This was an open-label, single center phase 1 study with 16 individuals. Eight healthy subjects [mean estimated glomerular filtration rate (eGFR) of 94.8 mL/min/1.73 m^2^] and eight patients with severe renal impairment (mean eGFR 21.9 mL/min/1.73 m^2^). Each one received a single dose of 50 mg of aprocitentan and was observed for 17 days. The pharmacokinetics of aprocitentan were similar in both groups with a T_max_ of 7.6 h in the SRFI group and 5.0 h among healthy subjects. Maximum plasma concentrations also did not differ. The half-life was longer in the SRFI group (53.2 h compared to 47.4 h in healthy individuals) due to reduced clearance. Aprocitentan was well tolerated in both groups [53].

Another phase 1 study recently completed the recruitment of 17 participants (healthy subjects matched to subjects with moderate hepatic impairment) with the intention of assessing the effect of moderate hepatic impairment due to cirrhosis on the pharmacokinetics of a single 25 mg dose of this agent (NCT04252495). The results are still pending.

Interestingly, the co-administration of aprocitentan, 50 mg, once daily, has been evaluated among 19 healthy male subjects receiving a single 8 mg dose of midazolam. Aprocitentan did not alter the pharmacokinetics of midazolam and was well tolerated when administered simultaneously. Therefore, this drug does not significantly interact with CYP3A4 substrates and they can be safely co-administered without dose adjustments [58]. Aprocitentan combined with rosuvastatin is also well tolerated [59].

Of note, the bioequivalence of two different tablet formulations has recently been evaluated in healthy subjects enrolled in an open-label, randomized phase 1 study and the results are still awaited (NCT05196399).

Additionally, one more study (placebo- and moxifloxacin-controlled randomized phase 1) has attempted to evaluate the effect of multiple-dose administration of aprocitentan on the electrical activity of the heart in healthy subjects by recording the QT interval duration, and the results are pending (NCT04281342).

The evaluation of the dose-response of aprocitentan on diastolic BP in subjects with grade 1 and 2 essential hypertension was conducted by a randomized, double-blind, multi-center and active comparator-placebo-controlled study [60]. Patients were randomly assigned to either placebo, aprocitentan 5 mg, 10 mg, 25 mg, or 50 mg, or lisinopril 20 mg daily. Participants’ 24 h ambulatory BP was recorded at baseline and week 8. Aprocitentan 10, 25, and 50 mg significantly decreased sitting systolic/ diastolic unattended automated office BP (placebo-corrected decreases: 7.05/4.93, 9.90/6.99 and 7.58/4.95 mmHg, respectively, *p* ≤ 0.014 versus placebo), compared with an unattended automated office BP reduction of 4.84/3.81 mmHg with lisinopril 20 mg. Aprocitentan was generally well tolerated in any of the dosage groups. Incidence of adverse events was similar in the aprocitentan groups (22.0% to 40.2%) and the placebo group (36.6%). The most common events were hypertension, headache, and nasopharyngitis. The authors also reported slight dose-dependent decreases in hemoglobin, albumin, and uric acid, an increase in estimated plasma volume, but no alteration in weight versus placebo. For further and comparative details, see the Table 1.

### 5.4. Large, Randomized Controlled Studies

The PaRallEl-group, Phase 3 study with aproCItentan in Subjects with ResIstant HypertensiON (PRECISION) study is the phase 3, international, multi-center, blinded, randomized study which investigated aprocitentan for the treatment of patients with RH [61] [Table 1]. The study had three sequential treatment parts. Its purpose was to evaluate the BP lowering effect of aprocitentan at 12.5 mg and 25 mg (Part 1) and the durability of its effects in long-term treatment with 25 mg aprocitentan for a further 32 weeks (Part 2), followed by a 12-week placebo-controlled withdrawal period when patients were re-randomized to receive either 25 mg aprocitentan or placebo (Part 3). Recently, positive top-line results were published [62]. Aprocitentan significantly reduced BP when added to standardized combination background therapy for RH over 48 weeks of treatment. Specifically, in the first double-blind treatment period of 4 weeks, 730 patients were randomized to receive aprocitentan 12.5 mg (*n* = 243), 25 mg (*n* = 243), or placebo (*n* = 244) orally, once daily for 4 weeks. Treatment with aprocitentan 12.5 mg and 25 mg met the primary endpoint and a statistically and clinically significant reduction in the unattended, automated office systolic BP was reported in both aprocitentan groups compared with placebo (a difference versus placebo of –3.8 [1.3] mm Hg for the dose of 12.5 mg [97.5% CI −6.8 to −0.8, *p* = 0.0042] and −3.7 [1.3] mm Hg for the dose of 25 mg [−6.7 to −0.8, *p* = 0.0046]). In Part 2 all patients received 25 mg of the drug once daily for 32 weeks and those who were already treated with aprocitentan during Part 1 maintained the mean reduction in systolic BP. During the following 12 weeks in Part 3, 614 patients were re-randomized to receive aprocitentan 25 mg or placebo and a significant increase in systolic BP was recorded for patients in the placebo group compared with those in the aprocitentan group (5.8 mm Hg, 95% CI 3.7 to 7.9, *p* < 0.0001) [Figure 2]. Therefore, the investigators concluded that aprocitentan reduces BP compared to placebo by week 4 of treatment and the effect is maintained over a period of 48 weeks. In general, aprocitentan was well tolerated. The most frequent adverse event was mild-to-moderate edema or fluid retention occurring in 9%, 18% and 2% of patients randomized to aprocitentan 12.5 mg, 25 mg, and the placebo group, respectively, mainly during the first 4 weeks of treatment.

## 6. Perspectives

Dual endothelin receptor antagonists lower BP and seem to offer a novel treatment option for BP control for patients whose hypertension is not adequately controlled despite the use of at least three other classes of antihypertensives, including a diuretic. However, the major issue is to lessen and avoid any of their side effects. The findings from both experimental and clinical studies with aprocitentan support its favorable tolerability, safety, and efficacy in lowering BP in patients with RH, achieving significant changes in BP within 14 days [60]. In addition, this agent enhances the antihypertensive effects of other drugs, such as renin-angiotensin-system (RAS) blockers [55], and its BP lowering effects have also been evaluated using ambulatory blood pressure monitoring (ABPM) [60], which is a more reliable predictor of cardiovascular outcomes and organ damage versus the office measurements [63,64].

Currently, although the BP reductions achieved with aprocitentan [60] have been identified as a surrogate for a reduction in cardiovascular morbidity [64,65], there is a considerable need for more and larger studies to extensively investigate its impact on major cardiovascular and renal endpoints in patients with RH and/or renal impairment. It would also be useful to further evaluate its impact on renal vascular resistance and left ventricular hypertrophy [55]. Ιn addition, there is a need to examine drug interactions with other antihypertensives [Table 2].

Overall, experimental and pharmacokinetic/pharmacodynamic studies provide promising findings, which, coupled with the findings from large RCTs, point towards significant benefits (in terms of blood pressure reduction) of aprocitentan in patients with RH. Several further steps are required, and many critical points should be clarified in order to elucidate the role of dual ET receptor antagonists in the therapeutic management of RH. For instance, it should be investigated whether the BP reduction is maintained for over a year, and the long-term safety of aprocitentan. The effect of the drug on target organ damage should also be examined. Also, it would be of great importance to investigate whether the BP reduction achieved with aprocitentan can be translated into cardiovascular benefits. Additionally, it would be useful for clinical practitioners to know if aprocitentan is better, similar, or worse than spironolactone as the fourth drug in the RH management algorithm, as well as if there are any subsets of patients that are more likely to be either responders or non-responders to aprocitentan.

## 7. Conclusions

Given the important role of the ET system in the cardiovascular system, the potential role that many ET antagonists have in the therapy of cardiovascular diseases, and the results from the studies published so far, aprocitentan has potential for the treatment of the subset of 12–15% of hypertensive patients who have RH. However, further large studies are needed to strengthen and establish its actions. The recent publication of the large PRECISION study in *The Lancet* journal provides further reassurance of the efficacy and safety of aprocitentan for RH, with the aim overcoming unmet needs in the management of this difficult group of patients.

## Figures and Tables

**Figure 1 life-13-00806-f001:**
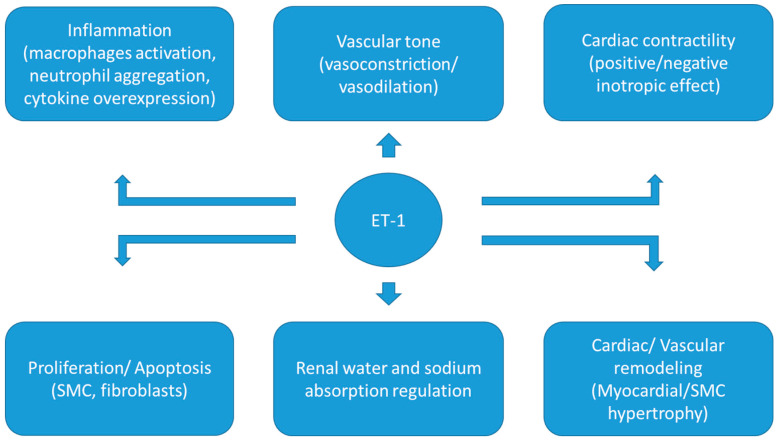
Endothelin-1: Effects on the cardiovascular system. SMC: Smooth Muscle Cells, ET-1: Endothelin-1.

**Figure 2 life-13-00806-f002:**
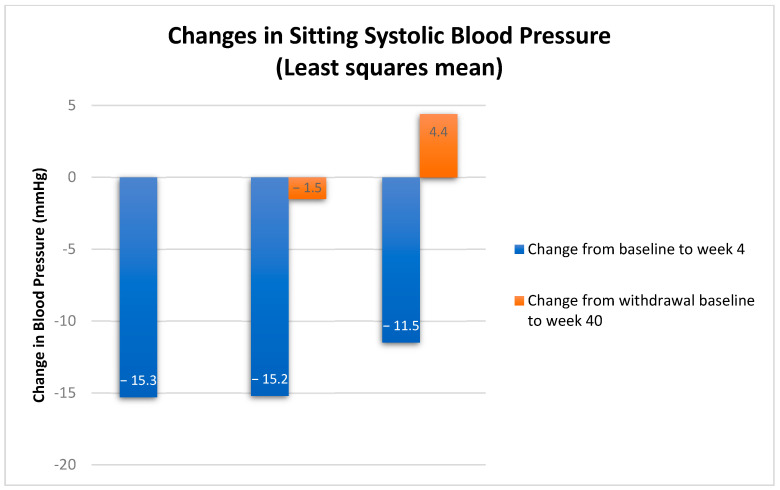
Changes in systolic blood pressure. Adapted from Slaich M.P., et al., *The Lancet* 2022 [62].

**Table 1 life-13-00806-t001:** Small and large clinical studies on aprocitentan.

	Participants/Groups	Aprocitentan Dosage	*n*	Duration	Results/Conclusions
Small clinical studies					
Sidharta et al., 2019 [52]	Healthy male and female subjects	600 mg (single doses) and 100 mg once a day (qd; multiple doses)	38 (Parts A&B) 34 (PartC)	30–33 days	Aprocitentan was well tolerated in healthy subjects with a pharmacokinetic profile favorable for qd dosing.
Fontes et al., 2020 [54]	Healthy Japanese and Caucasian male and female subjects	25 mg	20	10 days	No clinically relevant differences were found between Japanese and Caucasian subjects. Aprocitentan can be administered at a dose level of up to 25 mg in any ethnicity without dose adjustment.
Gueneau de Mussy, et al., 2021 [57]	Healthy subjects on a high sodium diet	10, 25, and 50 mg	23	9 days	Aprocitentan induced a moderate increase in body weight without obvious sodium retention
Sidharta et al., 2019 [53]	8 healthy subjects and 8 patients with severe renal function impairment (SRFI)	50 mg	16	17 days	The pharmacokinetics of aprocitentan were similar in both groups. Maximum plasma concentrations did not differ. The half-life was longer in the SRFI group due to reduced clearance.
NCT04252495	Healthy subjects matched to subjects with moderate hepatic impairment	25 mg	17	14 days	Pending (To assess the effect of moderate hepatic impairment on the pharmacokinetics of a single dose of aprocitentan 25 mg)
Sidharta et al., 2019 [58]	Healthy male subjects receiving 8 mg midazolam	50 mg	19	16–18 days	Aprocitentan did not alter the pharmacokinetics of midazolam and was well tolerated when administered simultaneously
Sidharta et al., 2020 [59]	Healthy males received a single dose of 10 mg rosuvastatin on days 1 and 13	25 mg	17	17 days	Aprocitentan does not affect the pharmacokinetics of rosuvastatin.
NCT05196399	Healthy subjects	25 mg	36	10 days	Pending (To evaluate the bioequivalence of two different tablet formulations)
NCT04281342	Healthy subjects receiving placebo or moxifloxacin	Multiple-dose administration	48	18 days	Pending (To evaluate the effect of multiple-dose administration of aprocitentan on the electrical activity of the heart)
Verweij, et al., 2020 [60]	Subjects with grade 1 and 2 essential hypertension receiving placebo, aprocitentan, or lisinopril 20 mg daily	5, 10, 25, or 50 mg	409	8 weeks	Aprocitentan 10, 25, and 50 mg significantly decreased sitting systolic/ diastolic unattended automated office BP, compared with the BP reduction with lisinopril 20 mg. Aprocitentan was generally well tolerated in any of the dose groups.
Large, randomized controlled study					
Danaietash, et al., 2022 [61], Schlaich 2022 [62]	Patients with RH	12.5 and 25 mg	730	4 weeks + 32 weeks + 12 weeks	To evaluate the BP lowering effect of aprocitentan at 12.5 mg and 25 mg (Part 1) and the durability of its effects with 25 mg aprocitentan for a further 32 weeks (Part 2), followed by a 12-week placebo-controlled withdrawal period (Part 3) Recent results: Aprocitentan reduces BP compared to placebo by week 4 of treatment and the effect is maintained over a period of 48 weeks.

**Table 2 life-13-00806-t002:** Drug-drug interactions between aprocitentan and other drugs.

	Drug	Subjects/Animals	Result
Trensz, et al., 2019 [55]	Valsartan and enalapril	Low-renin and normal-renin model of experimental hypertension (rats)	Synergistic in decreasing blood pressure in both experimental models
Sidharta, et al., 2020 [59]	Rosuvastatin	Healthy male subjects received a single dose of 10 mg rosuvastatin on days 1 and 13	Aprocitentan did not significantly affect the pharmacokinetics of rosuvastatin The combination of rosuvastatin and aprocitentan was well tolerated
Sidharta, et al., 2019 [58]	Midazolam	Healthy male subjects receiving 8 mg midazolam	Aprocitentan did not affect the area under the plasma concentration–time curve and the maximum plasma concentration of midazolam. There were no differences in tolerability parameters between treatments.

## Data Availability

No new data were created or analyzed in this study. Data sharing is not applicable to this article.

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
