# Peer review of "Novel Dual Endothelin Inhibitors in the Management of Resistant Hypertension"

_life, 2023, doi:10.3390/life13030806_

Round 1

Reviewer 1 Report

The review systematically discusses information about the various therapeutic options for RH. The review comprised important updates and discussion about prospective approaches in RH. I found this review suitable for the journal. However, there are some concerns as mentioned below. Addressing these comments will further enhance the quality of the review.

Comments:

1.     The treatment and prognosis of resistant hypertension due to secondary causes such as obstructive sleep apnea (OSA), primary aldosteronism, and renal artery stenosis.is crucial. Does identifying the secondary cause of RH play crucial role in the intervention. Please discuss. How is the treatment of secondary causes important.

2.     RH manifestation is associated with male sex, older age, black race, obesity, diabetes mellitus and CKD. Please provide a supporting reference.

3.     Previous study has systematically reviewed the Resistant hypertension. The authors should discuss this in their review. For instance, an increasing body of evidence has suggested benefits of mineralocorticoid receptor antagonists, such as eplerenone and spironolactone, in improving blood pressure control in patients with resistant hypertension, regardless of circulating aldosterone levels. Thus, this class of drugs should be considered for patients whose blood pressure remains elevated after treatment with a 3-drug regimen to maximal or near maximal doses. Vongpatanasin W. Resistant hypertension: a review of diagnosis and management. JAMA. 2014 Jun 4;311(21):2216-24. doi: 10.1001/jama.2014.5180. Erratum in: JAMA. 2014 Sep 17;312(11):1157. Dosage error in article text. PMID: 24893089.

4.     Recent updates from the group in the reference number 7 is already available. Endothelin: 30 Years from Discovery to Therapy. Please see discuss the updated information.

5.     The study from recent research by Liu et al should be discussed in the perspective of the pulmonary hypertension and ET antagonist. Liu C, Chen J, Gao Y, Deng B, Liu K. Endothelin receptor antagonists for pulmonary arterial hypertension. Cochrane Database Syst Rev. 2021 Mar 26;3(3):CD004434. doi: 10.1002/14651858.CD004434.pub6. PMID: 33765691; PMCID: PMC8094512.

Author Response

  1. The treatment and prognosis of resistant hypertension due to secondary causes such as obstructive sleep apnea (OSA), primary aldosteronism, and renal artery stenosis.is crucial. Does identifying the secondary cause of RH play crucial role in the intervention. Please discuss. How is the treatment of secondary causes important.

Thank you for the suggestion. We added this in the 2nd paragraph of our manuscript: “A crucial parameter of the management of …. and depends upon the specific disorder [6].”

  1. RH manifestation is associated with male sex, older age, black race, obesity, diabetes mellitus and CKD. Please provide a supporting reference.

Your comment is very useful. We added two appropriate references:

De La Sierra, A.; Banegas, J.R.; Oliveras, A.; Gorostidi, M.; Segura, J.; De La Cruz, J.J.; Armario, P.; Ruilope, L.M. Clinical Differences between Resistant Hypertensives and Patients Treated and Controlled with Three or Less Drugs. J. Hypertens. 2012, 30, 1211–1216, doi:10.1097/HJH.0B013E328353634E.

Braam, B.; Taler, S.J.; Rahman, M.; Fillaus, J.A.; Greco, B.A.; Forman, J.P.; Reisin, E.; Cohen, D.L.; Saklayen, M.G.; Hedayati, S.S. Recognition and Management of Resistant Hypertension. Clin. J. Am. Soc. Nephrol. 2017, 12, 524–535, doi:10.2215/CJN.06180616.

  1. Previous study has systematically reviewed the Resistant hypertension. The authors should discuss this in their review. For instance, an increasing body of evidence has suggested benefits of mineralocorticoid receptor antagonists, such as eplerenone and spironolactone, in improving blood pressure control in patients with resistant hypertension, regardless of circulating aldosterone levels. Thus, this class of drugs should be considered for patients whose blood pressure remains elevated after treatment with a 3-drug regimen to maximal or near maximal doses. Vongpatanasin W. Resistant hypertension: a review of diagnosis and management. JAMA. 2014 Jun 4;311(21):2216-24. doi: 10.1001/jama.2014.5180. Erratum in: JAMA. 2014 Sep 17;312(11):1157. Dosage error in article text. PMID: 24893089.

Thank you for your suggestion. We added this to the text (paragraph 1, section 2): “Furthermore, there is a growing body of evidence showing …. drugs to maximal or near maximal doses [6]”.

  1. Recent updates from the group in the reference number 7 is already available. Endothelin: 30 Years from Discovery to Therapy. Please see discuss the updated information.

Thank you for the correction. We added the recent, updated reference and changed the sentence appropriately.

  1. The study from recent research by Liu et al should be discussed in the perspective of the pulmonary hypertension and ET antagonist. Liu C, Chen J, Gao Y, Deng B, Liu K. Endothelin receptor antagonists for pulmonary arterial hypertension. Cochrane Database Syst Rev. 2021 Mar 26;3(3):CD004434. doi: 10.1002/14651858.CD004434.pub6. PMID: 33765691; PMCID: PMC8094512.

Thank you for your suggestion. We added this to the text (in paragraph 1, section 4): “It has been suggested that these agents increase ….. in patients with pulmonary hypertension compared with placebo [43].”

Reviewer 2 Report

In this review article, the authors present recent promising management of BP regards the ET-1 receptor antagonists in RH. The efficacy of Aprocitentan in patients with RH has been under investigation throughout recent studies as the authors mentioned in the text. comments are below

1. can you make a comparison table between small and large studies? it would be easy to catch up for the reader. 

2. can you provide a table of drug-drug interactions between aprocitentan and anti-hypertensive drugs depending on dose usage?  

3. Would you like to describe how ET-1 effect inflammation links to RH?

Author Response

  1. can you make a comparison table between small and large studies? it would be easy to catch up for the reader. 

This is a really good idea. We made and added the Table 1 to the text.

  1. can you provide a table of drug-drug interactions between aprocitentan and anti-hypertensive drugs depending on dose usage?  

Thank you for the suggestion. We added the Table 2. However, we included drug-drug interactions with any other drug, even not anti-hypertensive (valsartan, enalapril, midazolam, rosuvastatin), because there were only two anti-hypertensive drugs been studied. We also highlighted that in the text (paragraph 2, section 6): “Ιn addition, there is a need to examine drug interactions with other antihypertensives [Table 2].”

  1. Would you like to describe how ET-1 effect inflammation links to RH?

Yes, we have already described that in paragraph 6 of the third section in the manuscript. However, we added this sentence and its reference: “Therefore, the inflammation caused by ET-1 results both in atherosclerosis and myocardial hypertrophy, which contribute to RH.”

Reviewer 3 Report

Thank you for asking me to review this manuscript. This is an interesting review, whose aim is to present recent data on the development of dual ET inhibitors and on their use in resistant hypertension.

These drugs are currently being investigating and their roles in cardiovascular diseases is worthy of attention. Their importance is not limited to pulmonary hypertension, but they have a lot of potential cardiovasculare effect, as the authors mentioned.

The manuscript is well written and data are properly presented.

These are my minor issues:

-         Use the same font for the entire paper: in the abstract and in conclusion there are different types of font;

-         Could you make lines 53-54-55 (the identification of the role… inhibitors) clearer?

Author Response

-         Use the same font for the entire paper: in the abstract and in conclusion there are different types of font;

Dear Reviewer, we checked these sections of our manuscript, but they have the same font in our file.

-         Could you make lines 53-54-55 (the identification of the role… inhibitors) clearer?

Thank you for your suggestion. We changed that: “The identification of the role of endothelin (ET) system in vascular tone and blood pressure increase and its potential involvement in several cardiovascular diseases resulted in the development of ET receptor inhibitors and, subsequently, in the beginning of a new era in pharmacology”

Reviewer 4 Report

It is a lovely review. We slog on in this patient's group. We know that ET blockers are used in Pulmonary Arterial Hypertension; these drugs can be a new choice in this group. I have a suggestion for this paper. You can add some animal trials and especially emphasize the side effects of bosentan, macitentan, and ambrisentan in the pulmonary arterial hypertension group. 

Author Response

Thank you for your comment. We added this paragraph in the begging of the section 4: “Endothelin receptor antagonists, bosentan, ambrisentan…. and large blood vessels.”

Reviewer 5 Report

Dear authors

I have no spesific issues with the paper.

Congrats.

Author Response

Thank you so much!